# Hyperspectral imaging of microwave metasurfaces with deeply subwavelength resolution

Harry Penketh [1] ✉, Cameron P. Gallagher [1], Michal Mrnka [1], Christopher R. Lawrence[2], David B. Phillips [1], Ian R. Hooper [1] & Euan Hendry[1]

Electromagnetic metasurfaces offer a route to exotic, customised material properties that are not found in nature. However, in practice these artificial materials often do not live up to the promise of their design, limited by a myriad of fabrication challenges and defects. Global responses, such as transmission or reflection spectra, cannot distinguish different defect types, while scanning point measurements are impractical for materials which can contain thousands of individual meta-atoms. In this work we introduce a diagnostic imaging approach applicable to metasurfaces across the microwave, millimeter wave and THz bands, which we demonstrate with a microwave single-pixel camera. Using a near-field photomodulator, our approach can discern the resonance frequencies of individual meta-atoms in a complex microwave metasurface over large areas (80 x 80 mm), with spatial resolution far below the microwave wavelength ($\lambda$) and with capabilities beyond $\lambda/600$ at 15 GHz. We demonstrate high throughput visualisation of inhomogeneous broadening across various samples, while resolving near-field distributions of deeply subwavelength meta-atoms, and gaining valuable insight into the operational limitations of real-world metasurfaces.

Metamaterials are composite media that can be engineered to exhibit unique electromagnetic properties. Normally made up from subwavelength building blocks referred to as meta-atoms, they allow for extraordinary control over electromagnetic fields. Metasurfaces are the 2D equivalent: artificial sheet materials which can be lightweight, easier to fabricate, and can control wave propagation both on the surface and in the surrounding free space. They can be designed to enable beam shaping[1–5] in both transmission and reflection, polarise[6] or modulate[7,8] radiation, to act as a lens[5], or enable imaging[9–11]. However, their first and still highly relevant application is as a frequency-selective surface for filtering radiation[12–15]. As both the complexity and appetite for large-scale manufacture of metamaterials increases with time, so too will the need for more powerful methods for non-destructive evaluation of their performance[16].

Metasurfaces are usually composed of metallic or dielectric resonators arranged in planar or multi-layer configurations with sub-wavelength thickness, with the desired effect on radiation attained by the cumulative effect of many meta-atoms acting in sync. However, the interaction can deviate from design if the individual resonators do not function as desired. For example, imperfections during fabrication often cause alteration of dimensions or material properties, which can lead to degradation in performance. This lack of tolerance can become even more critical in layered metasurfaces, where interactions between layers are also crucial[17–20]. Such fabrication imperfections can have many unintended consequences such as spectral broadening, very often a problem for short wavelength (visible and IR) metasurfaces[21], where structures may require nanometric precision. However, discrepancies between modelling and experiment can arise

[1]Department of Physics and Astronomy, University of Exeter, Stocker Road, Exeter EX4 4QL, UK. [2]QinetiQ, Cody Technology Park, Ively Road, Farnborough GU14 0LX, UK. ✉e-mail: h.penketh2@exeter.ac.uk

due to inhomogeneities even at longer wavelengths[22], and can crucially impact device performance.

Figure 1a shows an example of this problem, comparing the measured total transmission of a layered dual-frequency band microwave metasurface with modelling predictions. Significant discrepancies are obvious: one observes a considerable reduction in peak transmission, shifted resonances and spectral broadening in the experimental data when compared to the "idealised" model. However, without the power of spatial discrimination, conclusively determining the root causes of such poor device performance may not be possible. Potential issues might include: fabrication tolerances, material permittivity uncertainties, surface roughness[23], alignment issues and localised defects. Understanding and mitigating the causes of these experimental discrepancies can be a challenging and time consuming process.

Any characterisation technique interrogates one specific aspect of a device. Microscopy can provide valuable information about the physical dimensions of components of a sample, with approaches such as atomic force microscopy and scanning electron microscopy offering the necessary resolution for high-frequency metamaterials[9,24–28]. However, the dimensional aspects of structures such as resonators and antennas do not necessarily map directly to their resonance frequencies. These also depend on local material properties and in layered structures may be highly sensitive to alignment, as shown in Fig. 1b.

In contrast, near-field scanning techniques may probe directly the electromagnetic response of a device within its operational frequency band[29,30], a crucial ingredient for discriminating between the factors suspected of causing the subpar device performance. Yet care must be taken to decouple the effect of the near-field probes on the system they are examining[31]. Most importantly, as these scanning approaches must physically move a probe on a point to point basis, they are therefore inherently slow. This slow scanning speed makes near-field scanning impractical for interrogating surfaces which contain thousands to millions of individual meta-atoms with high resolution. This limitation is not present for all-optical probing strategies[32,33]. Ideally, one would like to rapidly map spatial inhomogeneity to directly measure the response of individual resonators. However, this is typically very difficult to achieve due to their intrinsically subwavelength size. As a result, inhomogeneous broadening is regularly acknowledged but rarely precisely quantified.

In this paper, we present an imaging technique which is capable of spatially resolving the frequency response of individual meta-atoms in a microwave metasurface. By combining a microwave single-pixel camera leveraging photoexcited masks with a narrow band, frequency tuneable source, we gain valuable new insights into the inner-workings of complex metasurfaces. We demonstrate an application of this technique to spatially diagnose the cause of (and subsequently reduce) the effects of inhomogeneous broadening in a frequency selective metasurface, and to image the near fields of individual meta-atoms. In addition we explore the unique spatio-spectral signatures of a range of fabrication defects and features across multiple samples and show that our approach may be applied to different types of metasurface.

## Method

### Metasurface design

The samples under investigation comprise a metasurface pattern in copper atop a dielectric substrate (iTera MT40), shown in the inset of Fig. 1a. The repeating design is formed from square slot-spiral resonators with two different sizes arranged in a checkerboard pattern. The resonance frequency for these spirals is related to the length along the slot spiral, as well as the spacing between turns. The copper side of this printed circuit board (PCB) metasurface layer is mounted on a photoactive silicon layer.

The gap size between turns is 0.2 mm for both spiral sizes and the initial (deeply subwavelength) square side lengths are 1.349 and 1.184 mm for the spirals with low and high frequency resonances respectively. The copper layer has a thickness of 18 µm and the dielectric substrate has a thickness of 0.51 mm and a relative permittivity of $\epsilon_r = 3.45 + 0.011i$, as stated in the iTera MT40 datasheet. A detailed description of the metasurface geometry and associated COMSOL Multiphysics modelling can be found in Supplementary Information 1 Section 3. The layered metasurface is assembled as shown in the inset of Fig. 1a with the use of an external type vacuum sealer unit. Where cleaning of the mating surfaces is performed,

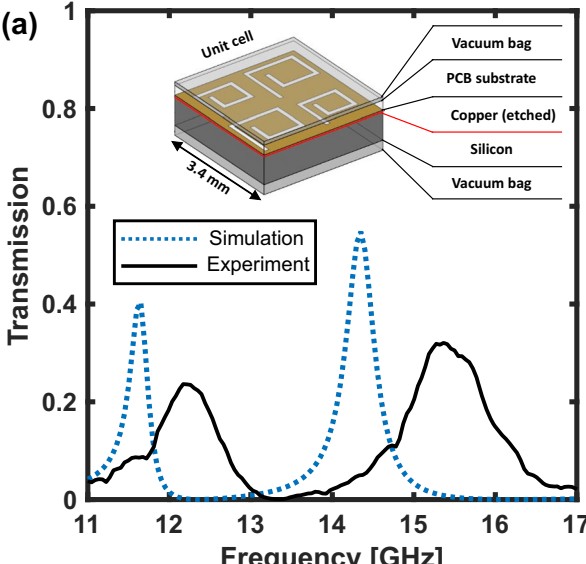

**(a)**

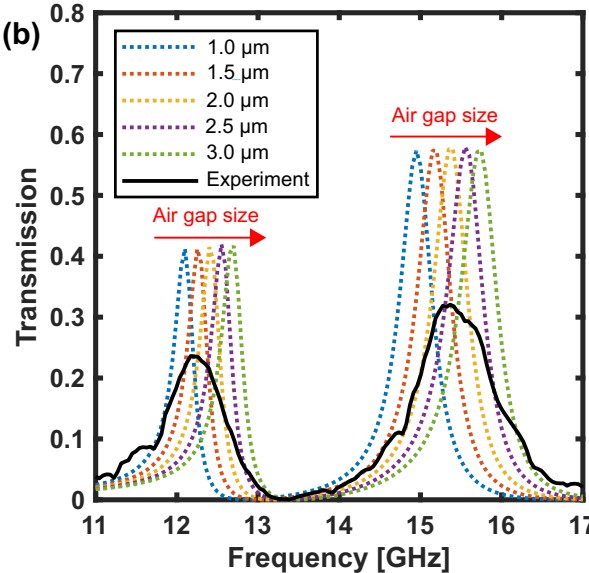

**(b)**

**Fig. 1 | Metasurface global transmission spectra. a** Experimental transmission spectrum (solid - black line) vs simulated spectrum from COMSOL model (dotted blue line), for the dual-band layered metasurface shown in the inset of the figure[43]. **b** Simulated effects of varying the size of an air gap between the silicon wafer and the metasurface layer (red region in inset of (**a**)).

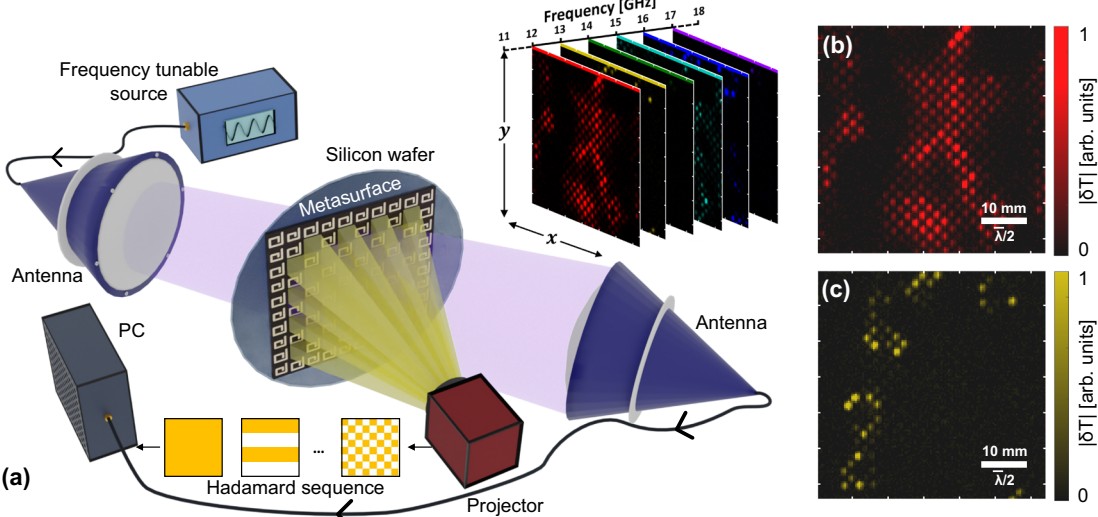

**Fig. 2 | Method for microwave single-pixel imaging.** In the experimental setup (**a**)[43], the microwave transmission through the metasurface is recorded using a frequency tuneable source and a pair of lens horn antennas whilst a sequence of optical patterns generates photoexcited masks within the silicon layer. Reconstructed images of the change in transmission upon local photoexcitation $|\delta T|$ are recorded over a range of frequencies, examples of which are shown in (**b**) and (**c**) at 12 GHz and 13 GHz respectively, for the metasurface shown in Fig. 1a. $\bar{\lambda}$ denotes the wavelength at 15 GHz in free space.

isopropanol is used in a 'drop and drag' method. The samples are not assembled in a cleanroom environment.

We initially choose a dual-band metasurface for the imaging tests presented in this work: as we will see below, this allows us to discern the two sub-lattices associated with the two different frequency bands as well as any anomalous resonators within each band. We record the global transverse-magnetic (TM) transmission spectrum of this metasurface at 35° to the surface normal with a vector network analyzer (VNA), as the solid line in Fig. 1a, b. As is often found for such samples, we observe clear discrepancies between this and the ideal model of this design (dashed line in Fig. 1a). It is difficult to immediately discern the origin of this discrepancy. For example, a distribution of element sizes due to spatial variation in the PCB etch rate could cause these effects, as the resonance frequency is correlated with the physical size of the elements. Similarly, inhomogeneity in the dielectric properties for the substrate material could potentially be to blame. There are also a number of more subtle candidates: in Fig. 1b, for example, we plot the simulated transmission spectra for metasurfaces with a small air gap between the copper and silicon layers. Here we see that an air gap of the order of microns in size significantly shifts the resonance frequencies, caused by a decrease in the effective permittivity around the resonators. A spatially varying air gap is therefore also a possible candidate for the origin of the inhomogeneous broadening observed in the spectrum of Fig. 1a.

## Imaging method

Microwave imaging of the metasurface is made possible by spatially modifying the photoactive silicon layer using visible light[32–36], an approach that is also suitable for millimeter wave and THz frequencies. Note that here we incorporate this photoactive layer into the metasurface design, but, as discussed and demonstrated later, the photoactive layer can instead be external to the metasurface. In the silicon layer (675 μm thick with high-resistivity), photoexcitation of the area around an individual meta-atom introduces electron-hole pairs, which provide a local loss channel due to the increase in local conductivity. Following refs. 37,38, we estimate that the resulting photocarrier density is ~$10^{18}/\text{m}^3$ for an excitation intensity of 100 W/m² and carrier lifetime of 30 μs as used here. Most importantly, the diffusion length of the photocarriers is ~200 μm[37]. This is much smaller than the spacing of the antennas, allowing us to modify selectively the active regions of

individual antennas (see Supplementary Information 1 Section 4 for a more detailed discussion of the image resolution).

The permittivity change of the silicon achieved upon photoexcitation (estimated to be ~-0.2i at 15 GHz—see Supplementary Information 1 Fig. S4), is calculated to yield a reduced transmission on resonance of around 40% (Supplementary Information 1 Section 5). As discussed in Supplementary Information 1 Section 5, we observe experimentally a lower modulation, an effect attributed in part to sample inhomogeneity. Nevertheless, we find this level of damping sufficient to selectively reduce the contributions of individual meta-atoms to the total metasurface response, allowing imaging with high signal-to-noise ratio. If one wanted to completely switch off the radiative coupling of a single meta-atom, a higher photoexcitation intensity would be required. Note also that the photocarrier lifetime can be engineered to increase the induced loss following[38], but this would come at the cost of reduced resolution through diffusion.

A schematic of the imaging system is shown in Fig. 2a. A pair of lens horn antennas are positioned to measure the TM microwave transmission at an angle of 35° to the normal of the metasurface shown in Fig. 1a, which includes a photoactive silicon wafer layer. The metasurface design exhibits minimal dispersion and so the choice of off normal illumination is for convenience of photoillumination. Unlike in the previous section where we described the measurements of global (i.e. of the entire sample) transmission spectra using a VNA, in these imaging experiments we utilise a separate source and detector for increased imaging speed. The microwave source is a frequency tuneable continuous wave signal generator which we sweep over the ~11–22 GHz range as indicated. We define the nominal wavelength as $\bar{\lambda} = 2$ cm at 15 GHz for subsequent imaging resolution calculations.

The transmitted power is detected by a coaxial Schottky-diode detector and transmitted to a PC DAQ card. The photomodulation sequence is provided by a projection system utilizing a digital micromirror device and a xenon arc lamp. An expanded description of the experimental setup, including demonstration of the visible pump and microwave beam uniformity, can be found in Supplementary Information 1 Section 2.

Our approach relies on the single-pixel imaging method (one that is intimately related to computational ghost imaging), in which the lack

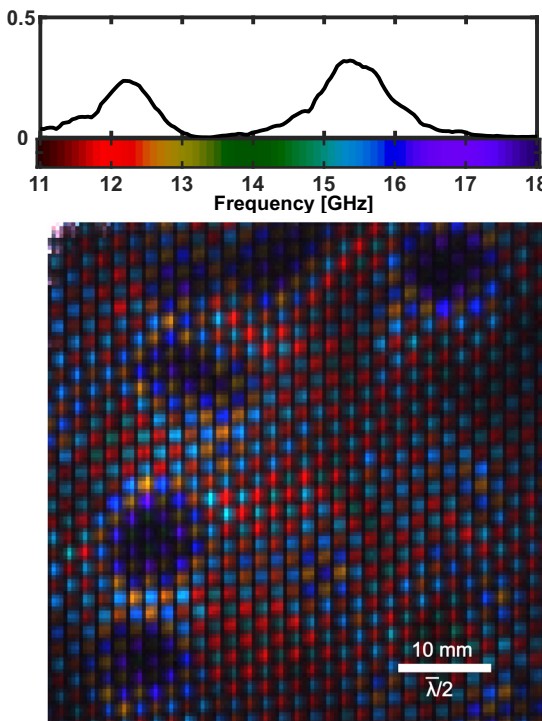

**Fig. 3 | False colour hyperspectral image of a region of the metasurface shown in Fig. 1a.** The global transmission spectrum is repeated inset for convenience. The colourmapping of microwave frequencies to visible colours is as shown inset, with the final image pixel colours given by the mixture of colours present in their local spectra. The FOV of the image is 54 × 54 mm at the metasurface. $\bar{\lambda}$ denotes the wavelength at 15 GHz in free space. The monochromatic images for the full dataset can be seen in Supplementary Information 1 Fig. S13.

of spatial resolution provided by a single 'bucket' detector is overcome by applying time-varying spatial modulations to the unknown light field[39,40]. By combining the known sequence of projections masking the metasurface, with the associated transmitted microwave signal measured by our diode detector, an image may be formed. Mathematically the process may be described as follows by reducing the dimensions of the object reconstruction **o** from $N \times N$ pixels to a vector of $N^2 \times 1$ pixels:

$$\mathbf{o} = \mathbf{P}^{-1}\mathbf{s}. \tag{1}$$

**P** is the illumination basis, a matrix with columns containing the vectorised 2D projections and **s** is the vector of bucket signals, measuring the spatial overlap between the projections and the object or field.

In the intuitive case where **P** describes the raster basis, i.e. a single 'on' pixel moving with each pattern, the detector signal vector **s** represents directly an image of the photoactivated change in transmission $|\delta T|$ at a given frequency, when reshaped to $N \times N$ pixels. Due to the limited photomodulation signal arising from one pixel (smaller than one meta-atom in this work), we opt for the more signal efficient Hadamard basis[41,42] shown in Fig. 2a.

The number of projections needed per image follows the number of pixels in the image as $2N^2$ (see Supplementary Information 1 Section 1 for more details). For the 128 × 128 pixel resolution images presented throughout this work, the number of projections per single-frequency image is 32,768, taking 1.6 s to acquire. We average 20 images to maximise signal-to-noise ratio and assemble hyperspectral data cubes of dimensions $128 \times 128 \times \Delta F/f$ points, where $\Delta F$ is the frequency range plotted (around 11–20 GHz) and $f$ the sampling interval of 100 MHz.

## Results

### Hyperspectral analysis

Figure 2b, c map the amplitude of the variation in transmission upon local photoexcitation, $|\delta T|$, at 12 GHz and 13 GHz, respectively. Each colour spot corresponds to the precise location of an individual meta-atom. One can immediately observe that different regions of the metasurface are active at different frequencies. At 12 GHz, corresponding to the lower frequency transmission peak in Fig. 1a, more of the surface is active. However, the signal across the imaging field of view (FOV) is far from homogeneous. Where one would expect a regular checkerboard pattern, large patches seemingly contain no active resonators at all. By examining the hyperspectral image dataset (shown in full in Supplementary Information 1 Fig. S13, but an example at 13 GHz is shown in Fig. 2c), we see that the images at nearby frequencies are complementary: i.e., the large dark regions in (b) are active at neighboring frequencies. It is clear then that we can both spatially and spectrally resolve the individual meta-atoms contributing to the overall inhomogeneous broadening seen in the total spectrum of Fig. 1a.

Taking the monochromatic images of Fig. 2b, c a step further, we can assemble a single image representing the full dataset with the spectral information encoded in the colour channels. We map the frequency range 11–18 GHz onto a colourmap based on the visible spectrum, and represent each pixel in the image with the weighted combination of colours (frequencies) present in its local spectrum, where brightness encodes relative signal. The resulting image is shown in Fig. 3, with the colourmap as indicated in Fig. 3 alongside a global transmission spectrum of the metasurface, as measured with a VNA. In Fig. 3 it can be seen that every meta-atom, across both the high and low frequency designed passbands, resonates at some frequency within the sampled range. Moreover, there is a clear pattern in regions which exhibit large frequency shift, indicated by the dark purple and orange colours: these resonators are grouped, and not randomly distributed across the surface. Furthermore, the two complementary sub-lattices, nominally corresponding to the colours red and light blue, undergo similar fractional frequency shifts in these regions.

These observations suggest that the inhomogeneous broadening observed here is not due to uncorrelated randomness in dimensional or material composition of the meta-atoms, but rather a systematic effect. This points us towards one of two likely causes. Firstly, the rate of PCB etching during fabrication can vary spatially, which can in turn result in meta-atom dimensions that vary slowly across the surface. Secondly, air gaps or contaminants between the layers in multilayered structures can substantially shift the resonance frequencies as shown in Fig. 1b, even when the gaps are as small as ~$\lambda$/10,000. Exploiting the rapid imaging capabilities of the system, we show in Supplementary Information 1 Fig. S9 that repeated assemblies of the layered structure vary the spatial distribution of these maximally blue-shifted regions. This suggests that interfacial contaminants, as opposed to fixed dimensional properties of the copper resonators, are impacting performance.

To test and verify this hypothesis, we introduce a series of controlled defects into one of our samples. In Fig. 4a, we take the metasurface from Fig. 3, clean it, and carefully lacerate a single meta-atom. Before reconstruction, we also introduce a single human hair (50 ± 10 μm thick), and a region covered by a very thin (10 ± 5 μm thick) PVC film. To incorporate all of these in one image, we extend the FOV to 80 × 80 mm, covering over 2000 meta-atoms, demonstrating the high throughput capabilities of our approach.

The region with the PVC film has an increased resonance frequency similar to the modelling results in Fig. 1b. Variations in the spectral shift in this region are attributed to the observed wrinkling of the deposited film. By contrast, the human hair induces very large spectral shifts that also vary rapidly in space, deforming around the region where the hair is present. The dark band corresponding to the

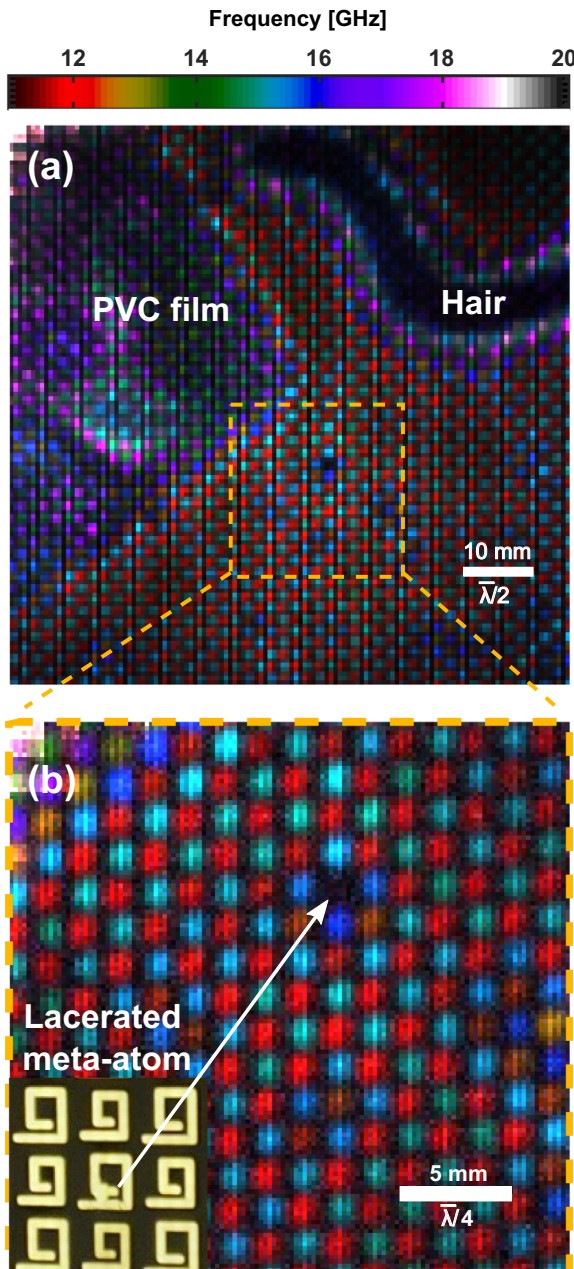

**Fig. 4 | False colour hyperspectral images of a sample with known defects.** The layered metasurface from Fig. 3, after cleaning, with a region of PVC film and a human hair added between layers and a single lacerated meta-atom, is shown with an 80 x 80 mm FOV in (**a**). In (**b**) the image is reacquired with reduced FOV of 25 x 25 mm providing enhanced resolution. $\bar{\lambda}$ denotes the wavelength at 15 GHz in free space. The monochromatic images for the full dataset of (**a**) and (**b**) can be seen in Supplementary Information 1 Fig. S15 and S16 respectively.

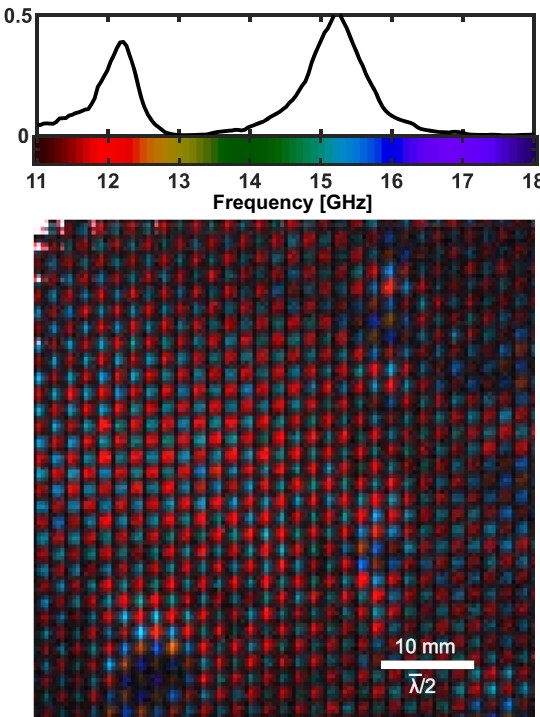

**Fig. 5 | False colour hyperspectral image as in Fig. 3, but with the sample replaced with one assembled with far fewer contaminants between layers.** The resulting much improved VNA transmission spectrum is shown inset. $\bar{\lambda}$ denotes the wavelength at 15 GHz in free space. The monochromatic images for the full dataset of can be seen in Supplementary Information 1 Fig. S14.

hair's location presents a substantial reduction in the total transmission of the metasurface across the imaged frequency band. Given such sensitivity to very subwavelength separations at the interface, even fine dust particles will degrade device performance significantly. This indicates the care which must be taken when assembling multilayer metamaterials, even at these long wavelengths.

The defective (lacerated) meta-atom appears as an isolated dark spot in Fig. 4a, as it no longer exhibits any resonant behaviour in this frequency band. By exploiting the ability of our DMD-based imaging approach to electronically reconfigure the imaging speed, FOV and resolution on-the-fly, we show in Fig. 4b that we can also

interrogate the fine details of the sample. With the imaging FOV reduced to just 25 × 25 mm (200 μm pixel size) we can observe clearly the behaviours of individual meta-atoms, making it straightforward to locate the faulty meta-atom. In addition, it allows us to resolve the subtle frequency shifts of elements neighbouring the defect. These possibly arise from modification of the element-element coupling in this region (a weak effect in this particular metasurface design) or a localised change in surface roughness upon laceration.

Based on the combined spatial and hyperspectral information revealed by our approach, we are able to identify with confidence the cause of the degraded device performance seen in Fig. 1a: the predominant defects are interfacial in nature. Equipped with this information, and finding an isopropanol clean alone does not completely remove these interfacial defects (see Supplementary Information 1 Fig. S9), we assemble a new cleaned section of metasurface with a pristine silicon wafer fresh from the distributor (note that this wafer has a different measured carrier lifetime of 55 μs, which slightly increases photocarrier diffusion and modulation efficiency). The resulting hyperspectral image and transmission spectrum are shown in Fig. 5. The reduced effect of inhomogeneous broadening is immediately clear from both the homogeneity of the image and the increased peak transmission in the global spectrum as measured with a VNA, i.e. we have vastly improved device performance, with peak transmission increasing by almost a factor of two.

The residual spectral offset between the simulated metasurface in Fig. 1a and the best experimental data in Fig. 5 must now be understood as an additional effect that is uniform across the sample, pointing towards a difference between the modelled and experimentally realised material properties or a spatially consistent geometric effect. Here, we posit that this is a result of the surface roughness of the copper of the PCB ($Rz \leq 2.5$ μm) introducing an effective intermediate air-gap layer that is not incorporated into our model.

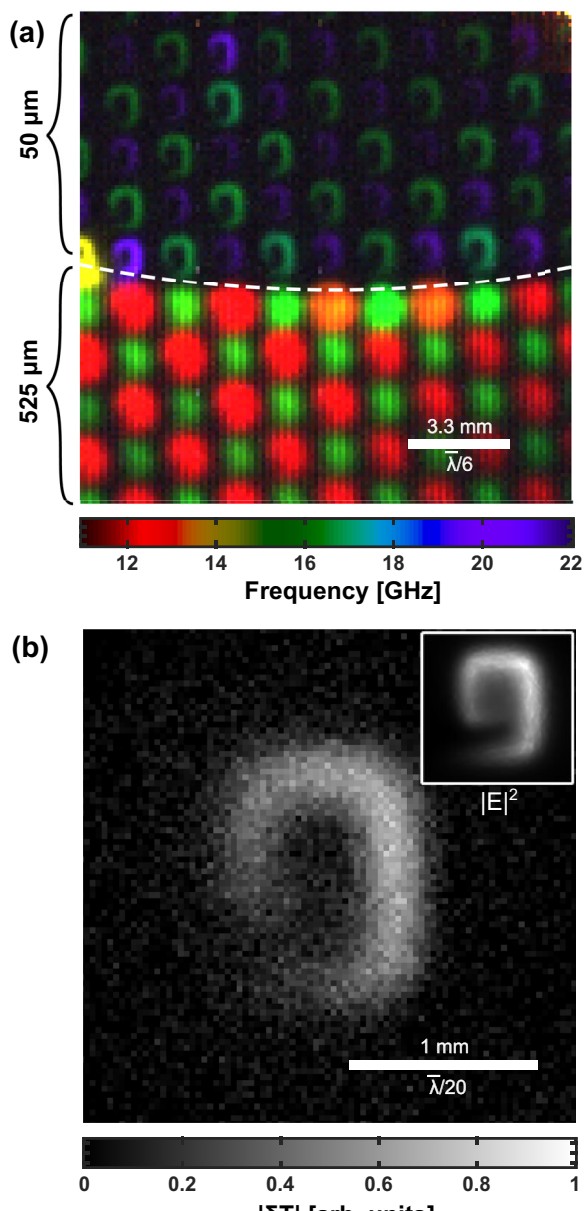

**Fig. 6 | Impact of photomodulator thickness on imaging. a** False colour hyperspectral image of a region of the metasurface shown in Fig. 1a, with the silicon layer replaced with a silicon membrane containing an outer ring 525 μm thick and an inner region 50 μm thick as indicated by the dashed white line. $\bar{\lambda}$ denotes the wavelength at 15 GHz in free space. The monochromatic images for the full dataset of (**a**) can be seen in Supplementary Information 1 Fig. S17. **b** A single meta-atom imaged at 18.5 GHz with an ultra-thin 6 μm membrane. In the inset of (**b**) the simulated (COMSOL) distribution of modulus squared electric field norm of one of the meta-atoms on resonance is shown for comparison. Further details can be found in Supplementary Information 1 Fig. S7.

In contrast to global spectral measurements, our approach reveals telltale spatio-spectral signatures of common fabrication defects and design features. Rectifying and decoupling effects such as these from those that vary spatially, such as the contaminants and defective meta-atom(s) imaged in this work, would present a considerable challenge without the ability to spatially resolve spectral features as shown here. This allows us to correctly identify point defects such as faulty meta-atoms in Fig. 4b and variance in meta-atom dimensions and interfacial contaminants in Figs. 3–5. Further examples including meta-atoms at region boundaries and variation in substrate thickness can be found

below in Fig. 6a, and for overetched areas of lithographically produced metasurfaces in Supplementary Information 1 Fig. S12.

## Discussion

For the metasurface presented above, the silicon layer forms an intrinsic part of the metasurface design. However, as we discuss in this section, our imaging approach may be applied to an arbitrary metasurface with minimal perturbation to the response of the surface under test. Imaging of a general, non-photoactive metasurface can be carried out by introducing an extrinsic silicon surface brought into close proximity. In introducing the photoactive layer, a deterministic global shift in resonance frequency is to be expected. Whilst some perturbation of the surface under test is inevitable in any near-field imaging technique, this single homogeneous shift is significantly easier to account for than the complex interaction with a scanning near-field probe for example. Nevertheless, as it is an aspiration of all near-field imaging approaches to be minimally perturbative, we explore two simple routes to achieving this below.

One can significantly reduce the perturbation by the silicon (indicated by the spectral shift of metasurface resonances due to the presence of the modulator) by reducing its spatial overlap with the evanescent near-fields of the meta-atoms (Supplementary Information 1 Fig. S10). A particularly effective means of doing so is to use a thinner silicon modulator, which will also enhance imaging resolution by reducing diffusion of the photomasks (Supplementary Information 1 Section 4). The trade-off is a reduction in carrier lifetime and therefore photomodulation efficiency, which we compensate for in the following demonstration with increased averaging (100 images) and illumination intensity (1000 W/m²).

In Fig. 6a we replace the original 675 μm silicon modulator with a thin 50 μm (~1 μs lifetime), 80 mm diameter membrane supported by a thicker 525 μm annulus from Silson Ltd. We show a hyperspectral image of a region of the annular transition between thick and thin silicon with a FOV of 16 × 16 mm. A clear transition is observed between the resonance frequencies above and below the annular threshold, with meta-atoms in the thin membrane region resonant closer to their expected, unloaded frequencies of 21 and 26 GHz. We even see the slight shifts expected in the resonant frequencies of meta-atoms near the interface between thin and thick modulator regions (marked by the white dotted line in Fig. 6a).

In Fig. 6a one can also directly observe the impact of the photomodulator thickness on the smearing of projected photomasks due to carrier diffusion, which affects imaging resolution (Supplementary Information 1 Section 4). In the thin membrane region, with resolution better than $\bar{\lambda}/150$, the detailed near-field structure of individual meta-atoms is revealed.

An alternative approach to reducing perturbation is to position the silicon modulator in the tails of the evanescent near fields of the meta-atoms. This strategy necessarily sacrifices photomodulation efficiency and imaging resolution. However, as the diffraction imposed resolution limit for a particular geometry is approximately given by the offset distance between the modulator and the metasurface[34], super-resolution imaging is achievable even for microwave metamaterials encased in dielectric layers of millimetric thickness. This approach is demonstrated in Supplementary Information 1 Fig. S12: hyperspectral imaging of a second metasurface (a simple array of dipole meta-atoms containing no intrinsic silicon), is performed with less than 1.5% shift in resonance frequency due to the silicon. A high photomodulation efficiency is maintained by increasing the charge carrier lifetime of the extrinsic modulator through surface passivation[38]. The resolution and contrast also remain high, revealing the dipolar near fields and defective regions caused by over etching which were previously concealed within the metasurface layers.

By choosing appropriately the modulator thickness, position and carrier lifetime, our imaging approach can be optimised for

application to an arbitrary metasurface. The practical limit for imaging resolution is given approximately by the modulator thickness (Supplementary Information 1 Section 4). In Fig. 6b we demonstrate imaging with a 6 μm thick silicon membrane (purchased from University Wafer), achieving a resolution better than $\bar{\lambda}/600$, limited only by the projection optics. This indicates capability for deeply sub-wavelength resolution beyond the microwave regime into millimeter wave and THz frequencies.

In summary this work demonstrates an imaging technique for characterising microwave, millimeter wave and THz metamaterials that is capable of mapping spectral inhomogeniety in metasurfaces with deeply subwavelength imaging resolution. By leveraging photo-induced conductivity in a photoactive silicon layer in close proximity to the resonators, we are able to selectively turn off (partially) the individual meta-atoms contributing to the metasurface response, and thereby identify the contribution of single meta-atoms to the global response of a complex metasurface.

As a tool for use in a manufacturing environment, the presented system offers high throughput, high contrast and high resolution non-destructive testing for metamaterials, phased arrays and antennas at modest cost. As a probe for fundamental research, this approach provides a platform for the in situ interrogation of phenomena which are inherently challenging to evaluate computationally, such as finite size effects, coupling between elements, inhomogeneous broadening and artefacts of the measurement apparatus.

## Data availability
The research data supporting this publication are openly available from https://doi.org/10.5281/zenodo.15208969.

## Code availability
The custom codes supporting this publication are openly available from https://doi.org/10.5281/zenodo.15208969.

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

## Acknowledgements

Engineering and Physical Sciences Research Council via EP/W003341/1 (H.P., C.P.G., I.R.H.), Engineering and Physical Sciences Research Council via EP/Y015673/1 (E.H., I.R.H), Engineering and Physical Sciences Research Council and QinetiQ Ltd. via EP/R004781/1 (H.P. and E.H.), Engineering and Physical Sciences Research Council via EP/S036466/1 (M.M.), European Research Council via 804626 (D.B.P), Royal Academy of Engineering (D.B.P) and Leverhulme trust (H.P.). For the purpose of open access, the author has applied a Creative Commons Attribution (CC BY) licence to any Author Accepted Manuscript version arising from this submission.

## Author contributions

H.P., M.M., and E.H. designed the microwave single-pixel imaging system and H.P. devised and implemented its application to metamaterial characterisation. H.P. performed the data analysis, and carried out the experiments with assistance from C.P.G. I.R.H. and C.P.G. designed and simulated the photoactive metasurface. D.B.P. and M.M. provided support on computational imaging aspects of the project. H.P. and E.H. wrote the manuscript. H.P., C.P.G., M.M., C.R.L., D.B.P., I.R.H., and E.H. all contributed to the interpretation of experimental data and edited the manuscript.

## Competing interests

The authors declare no competing interests.
