## [Transparent Peer Review file · Nature Communications]

Hyperspectral imaging of microwave metasurfaces with deeply subwavelength resolution

Corresponding Author: Dr Harry Penketh

Version 0:

Reviewer comments:

Reviewer #1

(Remarks to the Author)

The authors developed near-field single-pixel imaging system in the microwave regime and achieved hyperspectral imaging of microwave metasurfaces with deeply subwavelength resolution. They showed that the spatial resolution can reach around $\lambda/100$, and high throughput visualization of inhomogeneous broadening across various samples is also demonstrated. Although this work presents interesting and remarkable results, I believe the novelty does not meet the high standard required for publication in Nature Communications. My detailed concerns are as follows.

1. (Major) The motivation, as claimed by the authors, is to “solve this problem”. In the Abstract, “this problem” includes “global responses cannot distinguish different defect types”, which are further detailed with “physical dimensions”, “local material properties” and “alignment of layered structures” in the Introduction. However, after careful reading the manuscript, I did find any results showing that the authors really solved this problem. The problem solved, I believe, is the high-throughput visualization of inhomogeneous broadening across various samples with high spatial resolution, which is a problem faced by the point measurements, not the so-called global-responses measurements.
2. (Major) The setup is very similar to that in the terahertz regime, as admitted by the authors. To achieve near-field single-pixel imaging, a thin silicon wafer with small distance to the metasurface is required for high spatial resolution. Therefore, the application of this work is very limited: only metasurfaces fabricated directly on photoactive semiconductor (such as silicon) can be hyperspectrally imaged. For other metasurfaces fabricated on other substrates, the spatial resolution may be very limited or even the developed approach does not work. Therefore, I am worried on the broad interest of this work.
3. (Major) Recently, Stantchev et al demonstrated that $\lambda/45$ can be achieved at 0.75 THz by using an ultra-thin (6 μm) silicon wafer (Optica 2017, 4:989). The spatial resolution is mainly determined by the carrier diffusion. By scaling this to the microwave regime, I do not think $\lambda/100$ spatial resolution is a difficult task.
4. (Major) The authors did not provide data to show whether the photonic and microwave beams are uniform. Indeed, this is vital for the imaging reconstruction. Without the reference data showing the uniformity of the photonic and microwave beams, it is difficult to determine whether the dark regions are due to defects of the metasurfaces or defects of the illuminations.
5. (Major) The organization should be greatly improved. For example, Fig. 3(b) discussed long after Fig. 3(a).
6. (Minor) Fig. S2 is misleading. The thickness of silicon wafer is not plotted in scale.

There may exist other minor problems. But I think the major problems such as overclaims/incorrect claims, limited novelty and lack of broad interest, and insufficient experimental supporting data, suggest that this work does not meet the high criteria of publication in this journal. Therefore, I cannot recommend acceptance of this manuscript.

Reviewer #2

(Remarks to the Author)

The authors describe an all-optical characterization technique adapted for microwave metasurfaces. It enables visualization of the GHz resonance frequency of individual meta-atoms in an expedite manner. The authors show spatially and frequency resolved wide 80x80 mm images of a microwave metasurface.

The manuscript is well-written, and the authors present a rigorous scientific description of their experimental results, along with a thorough numerical and theoretical analysis.

However, one key aspect that could be improved is the clarity of the experimental setup. At first glance, it was not

immediately obvious that the projector is scanning the metasurface with different optical patterns. Figure 2 oversimplifies the setup, potentially misleading readers into thinking that only a few antennas and a projector are required. The critical process of rastering the surface with optimized optical patterns is underemphasized and should be more explicitly highlighted both visually in the figure and in the description on page 4. For instance, the Hadamard projection sequence, which is currently described in the supplementary material, should be at least partially incorporated into the main text to provide readers with a more complete understanding of this crucial aspect.

Beyond these suggestions to make the work more accessible to a broader audience, the paper is a comprehensive study, well-supported by simulations. My only concern lies with the relevance of the topic. While the work on microwave metasurfaces is robust, this area is not particularly novel, with much of the current research focus having shifted to metasurfaces in the visible range, which is a hotter topic.

I leave it to the Editor to decide whether this work, under these circumstances, warrants publication in Nature Communications.

Reviewer #3

(Remarks to the Author)

In this study, Harry et al. introduce a novel technique for characterizing the near-field electromagnetic properties of metamaterials. The experimental results underscore the method's high sensitivity and resolution, effectively unveiling the specific effects of various defects on the resonant states of meta-atoms. This approach offers an innovative and potent experimental tool for the assessment and analysis of metamaterials' electromagnetic characteristics.

Overall, my assessment is positive. However, I have some comments and suggestions where further clarification and improvement could enhance the manuscript:

1. Compared to the work in references [32-33], the introduction of single-pixel imaging methods is one of the innovative aspects of the work. However, the main text and supplementary materials do not provide a clear elucidation of the implementation methods and system structure for the employed single-pixel imaging technique. The authors are advised to provide a more detailed introduction, including additional textual descriptions, structural diagrams, photos of the actual setup, or experimental photos, etc.
2. Further details on the structure of the excitation light source system are suggested. Furthermore, how does the system ensure uniform light intensity across various measurement positions within the illuminated area? Additionally, could the structure of the lighting system potentially interfere with the electromagnetic field of the metamaterials under test?
3. What are the main considerations for the TM microwave transmission at an angle of 35° to the normal of the metasurface? Does this requirement restrict the applicability of this method to certain types of metamaterials, such as those sensitive to the angle of incidence?
4. Mapping results at different frequencies to the visible spectrum is a creative and intriguing method of display. However, as evident in Figure S9, the impact of varying frequency sampling is significant. Does the choice of frequency sampling interval affect the accuracy and reliability of diagnostic results across various meta-atom areas, as well as the complementary trends observed in images at nearby frequencies? Is there a foundational basis for selecting the frequency sampling intervals?

Version 1:

Reviewer comments:

Reviewer #1

(Remarks to the Author)

The authors satisfactorily addressed most of the concerns raised by the reviewers. There exists one remaining problem in the revised manuscript. "Spatial resolution beyond $\lambda/600$ " is a claim highlighted in the Abstract, however, the supporting data figure is provided in the Supplement Information. I strongly recommend the authors to move or copy the key figures to the main text to provide strong support to this important claim.

Reviewer #2

(Remarks to the Author)

With the recent revisions to the manuscript, I now have a clearer understanding of the broader impact of their work across the microwave, millimeter-wave, and THz bands. The study is sound, and I now believe that it deserves publication in Nature Communications.

Reviewer #3

(Remarks to the Author)

The authors refer to Fig. RS12(c)(d) in their response to R3.4, but these subfigures are not included in the Supplementary Material. However, all other concerns raised in my initial review have been adequately addressed, and I have no further comments to add.

Response letter

Hyperspectral imaging of microwave metasurfaces with deeply subwavelength resolution

Harry Penketh,^{1,*} Cameron P. Gallagher,¹ Michal Mrnka,¹ Christopher R. Lawrence,² David B. Phillips,¹ Ian R. Hooper,¹ and Euan Hendry¹

¹*Department of Physics and Astronomy, University of Exeter, Exeter, EX4 4QL, UK.*

²*QinetiQ, Cody Technology Park, Ively Road, Farnborough, GU14 0LX, UK*

We would like to thank the reviewers for their time and detailed feedback. In response we have expanded upon our manuscript with new experimental demonstrations and added considerable supporting data to the supplementary information. In particular, by clarifying and experimentally demonstrating the applicability of our approach to a general metasurface, we believe the manuscript is now significantly strengthened. We detail our responses to the reviewers' comments below in blue text.

The following notation is adopted to distinguish new from old figures:

M - the old submitted manuscript.

MS - the old submitted supplement 1.

R - the new revised manuscript.

RS - the new revised manuscript supplement 1.

L - this response letter.

REVIEWER COMMENTS:

REVIEWER 1

“The authors developed near-field single-pixel imaging system in the microwave regime and achieved hyperspectral imaging of microwave metasurfaces with deeply subwavelength resolution. They showed that the spatial resolution can reach around $\lambda/100$, and high throughput visualization of inhomogeneous broadening across various samples is also demonstrated. Although this work presents interesting and remarkable results, I believe the novelty does not meet the high standard required for publication in Nature Communications. My detailed concerns are as follows.”

Response: We thank the reviewer for their feedback, but wish to clarify that the novelty of our work does not rely on achieving the highest possible resolution. As we demonstrate below, we can achieve markedly higher imaging resolution in our experiment, but this increased resolution yields no additional information. Instead, the novelty lies in the high-throughput approach we use to spatially locate the origin of inhomogeneous broadening - to the best of our knowledge, the first time such an approach has been applied to a metasurface.

R1.1: “(Major) The motivation, as claimed by the authors, is to “solve this problem”. In the Abstract, “this problem” includes “global responses cannot distinguish different defect types”, which are further detailed with “physical dimensions”, “local material properties” and “alignment of layered structures” in the Introduction. However, after careful reading the manuscript, I did find any results showing that the authors really solved this problem. The problem solved, I believe, is the high-throughput visualization of inhomogeneous broadening across various samples with high spatial resolution, which is a problem faced by the point measurements, not the so-called global-responses measurements.”

Response: We strongly believe that “high-throughput visualization of inhomogeneous broadening across various samples with high spatial resolution” is the solution to this problem, and reassert that global measurements such as reflection and transmission spectra cannot distinguish different types of defect. In the example given in the main paper, we show that inhomogeneous broadening occurs due to dust particles trapped between layers of our

* h.penketh2@exeter.ac.uk

metasurface. From comparison between the global transmission spectra and finite element modeling we can see that some resonators are operating outside their designed frequency bandwidth, but we cannot differentiate where these resonators are positioned on the surface. They could be randomly distributed (as might be expected from fabrication defects), arranged in patches (as would be expected from over-etching) or, as it turns out, distributed in circular domains. It is this extra information, only available from our visualization approach, that points us to the correct diagnosis. Moreover, a simple image measured with one frequency does not clearly identify the position or origin of a defect (for example, compare the single frequency images from Fig. R2(b) to our visualization in Fig. R3). Only once the frequency response is combined in our hyperspectral maps can one properly identify defect type. This is evident in our ability to identify a range of different defect types across multiple samples: point defects such as faulty meta-atoms (in Fig. R4(b)) and variance in meta-atom dimensions (spiral lengths in Figs. R3-6), overetched areas of lithographically produced metasurfaces (Fig. RS12), meta-atoms at region boundaries and variation in substrate thickness (Fig. R6), and dust particles between layers of a metasurface (Fig. R3). All of these have characteristic information in hyperspectral maps which are incomplete when imaging with a singular frequency or measuring a global spectrum. We therefore respectfully reassert our claim that the presented method is able to distinguish different defect types, and agree that we did not expand on this sufficiently in the previous manuscript. We now better clarify this point at the end of the subsection ‘Hyperspectral analysis’, by identifying the variety of defects and features we have identified in this work and also referring to new measurements in both the revised manuscript and a considerably expanded supplementary information document:

“In contrast with global spectral measurements, our approach reveals telltale spatio-spectral signatures of common fabrication defects and design features. ... This allows us to correctly identify point defects such as faulty meta-atoms in Fig. R4(b) and variance in meta-atom dimensions and interfacial contaminants in Figures R3-R5. Further examples including meta-atoms at region boundaries and variation in substrate thickness can be found below in Fig. R6 and for overetched areas of lithographically produced metasurfaces in Fig. RS12.”

R1.2a:“(Major) The setup is very similar to that in the terahertz regime, as admitted by the authors. To achieve near-field single-pixel imaging, a thin silicon wafer with small distance to the metasurface is required for high spatial resolution. Therefore, the application of this work is very limited: only metasurfaces fabricated directly on photoactive semiconductor (such as silicon) can be hyperspectrally imaged.”

Response: We agree that it is important to demonstrate the broader applicability of our technique to general metasurfaces, which we now explicitly address in an entirely new portion of the discussion focused on minimally perturbative imaging of arbitrary metasurfaces (which is also a new dedicated section of the supplementary material). We show experimentally in Fig. RS12 that not only is it incorrect that “only metasurfaces fabricated directly on photoactive semiconductor (such as silicon) can be hyperspectrally imaged”, but that imaging can be performed with high resolution and minimal perturbation to the metasurface.

R1.2b:“ For other metasurfaces fabricated on other substrates, the spatial resolution may be very limited or even the developed approach does not work. Therefore, I am worried on the broad interest of this work.”

Response: We agree that resolution is an important consideration when applying our imaging approach to a general metasurface, though we point out there is an intrinsic trade off between resolution and imaging speed. As a rule of thumb, the resolution limit for a particular geometry is given by the offset distance (i.e. the distance between photomasks and the surface due to the finite thickness of the modulator and/or any encapsulating layers) [1]. We then tune the modulator lifetime and corresponding diffusion length (i.e. how far the photo-carriers travel in their average lifetime) so that the resolution limit is maintained. Modulator thickness and lifetime are adjustable parameters, and we have chosen these to target an imaging resolution better than the typical size of a meta-atom in our measured frequency range. However, for many of the common metasurface defect types there is no diagnostic requirement for very sub-wavelength resolution. While we can tune our modulator to improve resolution (Fig. RS7(L4) below is taken with $\sim \lambda/600$ resolution), such resolution does not provide any additional information while having significant overhead, typically requiring more intense light for modulation, smaller area and/or longer imaging times. Nevertheless, the tunability in resolution indicates that our approach could be adapted for higher (e.g. THz) frequencies.

For imaging metasurfaces specifically, perturbation (e.g. a shift in resonance frequency) is generally a far more important effect than achieving highly subwavelength resolution, as the modulator thickness and location also influence the response of the metasurface itself. Perturbation is a well known problem in metasurface imaging: for example, tips used for efficient near field scattering often perturb the very electromagnetic fields they are trying to characterize. As such, it is an aspiration of ALL near field imaging approaches to be as minimally perturbative as possible. In

FIG. 1. (a) Results of simulations of the dipolar metasurface geometry shown in Fig. RS10, with varying thicknesses of silicon photomodulator included as indicated. Square root of the normal incidence transmission as a function of frequency. (b) and (c), as in (a) but including an air gap between the metasurface and modulator of 100 and 200 μm respectively.

Fig. RS10(L1)(a), we see that one approach towards minimal perturbation is to reduce the modulator thickness. This is demonstrated through simulations of a dipolar metasurface that is loaded on one side by a variable thickness silicon layer (for more details see the new Supplement 1 Section 7). By reducing the total overlap of a meta-atom's near fields with the modulator layer, the resonance frequency remains much closer to the unloaded metasurface.

We can demonstrate this effect experimentally using large area silicon membranes. In Fig. R6(L2) [which replaces the original Fig. M5 in the previous manuscript], by replacing the 675 μm silicon layer with an ultra-thin 50 μm membrane supported by a thicker 525 μm annulus, we see a striking change in the resonant frequency of meta-atoms positioned above and below the annular threshold, with intermediate frequencies observed for some meta-atoms close to the border. In Fig. R6(L2) we can also directly observe the impact of the photomodulator thickness on the imaging resolution. As shown in Fig. RS4, reducing the thickness of the silicon reduces the carrier diffusion, which improves spatial resolution. The resulting $\sim \lambda/150$ resolution for the membrane region of Fig. R6(L2) shows the near fields of the meta-atoms in more detail alongside their spectral characteristics. In general, the thinner the membrane, the

FIG. 2. False colour hyperspectral image of a region of the metasurface shown in Fig. M1(a), with the silicon layer replaced with a silicon membrane containing an outer ring 525 μm thick and an inner region 50 μm thick as indicated by the white dashed line. The simulated electric field distribution of one of the meta-atoms on resonance is shown inset for comparison.

smaller the impact on the intrinsic functionality of the metasurface under investigation. A ‘soft’ limit on the thickness of the wafer is imposed by the absorption of the visible pump light. A 6 μm thick wafer will absorb over 85% of an incident red pump light (after reflection losses) following the parameters specified in Supplement 1 Section 3, and has been experimentally applied to near-field imaging [2]. To efficiently go thinner still, one could look to photo-excite above the direct band gap energy of silicon with wavelengths below ~ 350 nm, where comparable absorption may be achieved in tens of nm.

A second way to minimise perturbation is to place the modulator in the tails of the meta-atom evanescent fields - this is the approach typically used in tip scattering imaging approaches to reduce perturbation effects. Increasing the distance between the modulator and the meta-atoms is a simple but effective way to reduce the spectral shift introduced by its presence, as shown in Fig. RS10(L1(a-c)). However, this approach has important trade offs in terms of both modulation efficiency and imaging resolution. Fortunately, a reduced modulation efficiency can be compensated for by increasing the lifetime of the charge carriers in the silicon or the illumination intensity [3]. In Fig. RS12(L3) we demonstrate minimally perturbative imaging of a new, non-photoactive metasurface by means of introducing a 100 μm thick double-sided passivated silicon modulator with a carrier lifetime of 1.3 ms. The metasurface design is shown in the figure insets consisting of dipolar copper crosses with 4-fold symmetry. The unperturbed metasurface (dashed red box inset top left) supports a resonance at $\simeq 15.2$ GHz, shown in the reflection spectrum in (i). When loaded with the thin silicon modulator, positioned 280 μm away from the layer of metallic meta-atoms, the shift in resonance frequency is as small as 200 MHz, as shown in (i). Crucially, the hyperspectral image (a) still shows excellent contrast, throughput and resolution. One can clearly see the individual dipolar fields of the vertical metallic bars as well as defective regions caused by over etching of the copper during fabrication.

Finally, we would like to point out that the perturbative effects arising from our modulator (a single, homogeneous layer) are significantly easier to calculate accurately (and therefore mitigate) than for common near field imaging approaches e.g. employing tip scattering.

R1.3:“(Major) Recently, Stantchev et al demonstrated that $\lambda/45$ can be achieved at 0.75 THz by using an ultra-thin (6 μm) silicon wafer (Optica 2017, 4:989). The spatial resolution is mainly determined by the carrier diffusion. By scaling this to the microwave regime, I do not think $\lambda/100$ spatial resolution is a difficult task.”

Response: Achieving deeply sub-wavelength ($\sim \lambda/100$ to $\sim \lambda/10$) resolution is an important strength of our tech-

FIG. 3. (a) Hyperspectral image (TE, 35° reflection) of a simple array of crossed dipole meta-atoms containing no intrinsic silicon (copper rectangles as shown in lower inset of (a)). The $100\ \mu\text{m}$ thick passivated silicon modulator is introduced to the layered metasurface (red dashed box) as shown in the upper inset of (a) and then the structure is vacuum sealed for improved uniformity. In (i), VNA reflection spectra before and after the introduction of the silicon reveal that the spectral response of the metasurface is almost entirely unperturbed. The ‘thin PCB’ substrate in the figure inset is $0.28\ \text{mm}$ thick.

nique, as it facilitates interrogation of individual meta-atoms. However, whilst an extension to ‘ultra sub-wavelength’ ($\sim \lambda/1000$) resolution is undoubtedly possible (and demonstrates capabilities for high frequency metasurfaces), it represents a poor optimisation to the imaging task at hand. We include in Fig. L4 below (and new Fig. RS7 in the supplementary) a comparison between imaging of a single meta-atom using a thin $50\ \mu\text{m}$ silicon membrane and an ultra-thin $6\ \mu\text{m}$ membrane, as used in [2]. For the $6\ \mu\text{m}$ modulator in (b), the resolution is limited by the size of the projected pixels: for this demonstration we have adjusted the imaging optics to achieve a $30\ \mu\text{m}$ resolution. However, it is clear that, in this particular case, better than $\sim \lambda/600$ resolution yields no new insights - the field structure is the same in both cases - despite the significantly reduced imaging area of the $6\ \mu\text{m}$ wafer. This emphasizes, as discussed in the response above, the importance of tailoring resolution appropriate to the problem at hand.

R1.4:“(Major) The authors did not provide data to show whether the photonic and microwave beams are uniform. Indeed, this is vital for the imaging reconstruction. Without the reference data showing the uniformity of the photonic and microwave beams, it is difficult to determine whether the dark regions are due to defects of the metasurfaces or defects of the illuminations.”

Response: We again thank the reviewer for pointing this out. We now include images of the visible and microwave beams in the supplementary material in Fig. RS2 and refer these in the main text: *An expanded description*

FIG. 4. Comparison between single-frequency imaging of a single meta-atom with a 50 μm and 6 μm thick silicon modulator, at 16.3 GHz and 18.5 GHz respectively. Images cropped from a larger 128×128 pixel image. The number of images averaged (200) and illumination intensity (1000 W/m^2) are the same for both. The silicon membranes are shown in the figure insets.

of the experimental setup, including demonstration of the visible pump and microwave beam uniformity, can be found in Supplement 1 Section 2. The features we see in imaging are not due to any spatial variation in the beams themselves, as we can observe clear sample to sample variations (compare Fig. R3 with the almost homogeneous R5 for example). Furthermore, the localised signal maxima observed when imaging a metasurface are unambiguously arising due to enhanced interactions with the meta-atoms. The dark spots that the referee refers to are observed only in single frequency images, arising when groups of meta-atoms are simultaneously non-resonant. When mapped hyperspectrally, we see signal maxima accounting for each and every meta-atom, as in Fig. R5.

R1.5:“(Major) The organization should be greatly improved. For example, Fig. 3(b) discussed long after Fig. 3(a).”

Response: We have improved the organisation of the paper as suggested and reassigned the section headings. A new portion of the discussion section focused on minimally perturbative imaging of arbitrary metasurfaces has been added, the imaging method section expanded and adjustments to the narrative have been made accordingly. What was formerly Fig. M3(b), now Fig. R5, appears where it is discussed in the text.

R1.6:“(Minor) Fig. S2 is misleading. The thickness of silicon wafer is not plotted in scale.”

Response: We thank the reviewer for this observation and have now replotted Fig. RS4 to scale.

REVIEWER 2

“The authors describe an all-optical characterization technique adapted for microwave metasurfaces. It enables visualization of the GHz resonance frequency of individual meta-atoms in an expedite manner. The authors show spatially and frequency resolved wide 80×80 mm images of a microwave metasurface. The manuscript is well-written, and the authors present a rigorous scientific description of their experimental results, along with a thorough numerical and theoretical analysis.”

R2.1:“However, one key aspect that could be improved is the clarity of the experimental setup. At first glance, it was not immediately obvious that the projector is scanning the metasurface with different optical patterns. Figure 2 oversimplifies the setup, potentially misleading readers into thinking that only a few antennas and a projector are required. The critical process of rastering the surface with optimized optical patterns is underemphasized and should be more explicitly highlighted both visually in the figure and in the description on page 4. For instance, the Hadamard projection sequence, which is currently described in the supplementary material, should be at least partially incorporated into the main text to provide readers with a more complete understanding of this crucial aspect.”

Response:

We thank the reviewer for these recommendations to improve the clarity of our method. It is essentially true that “only a few antennas and a projector are required”, this is one of the strengths of this approach vs delicate mechanical

near-field scanning probes. The importance of the sequence of illuminations has now been emphasised, as suggested by the reviewer, as follows:

Fig. R2 has been modified to display some example Hadamard patterns and emphasize that a sequence of projections is needed. A full description of the single imaging process has been added to the main text, emphasising the need for multiple projections. Photographs of the experimental setup have been added to the supplementary material in Fig. RS1.

R2.2: “Beyond these suggestions to make the work more accessible to a broader audience, the paper is a comprehensive study, well-supported by simulations. My only concern lies with the relevance of the topic. While the work on microwave metasurfaces is robust, this area is not particularly novel, with much of the current research focus having shifted to metasurfaces in the visible range, which is a hotter topic. I leave it to the Editor to decide whether this work, under these circumstances, warrants publication in Nature Communications.”

Response:

We thank the reviewer for their comments on the completeness of our work. However, we disagree with the suggestion that microwave metasurfaces and metamaterials are not important. Microwave metamaterial absorbers [4, 5], time-varying microwave metamaterials [6–9], microwave metamaterial BICs [10, 11] and microwave Huygens metamaterials [12], are just some examples of major breakthroughs in this frequency region resulting in high impact publications in the last few years. Furthermore, it is at these lower frequencies that metamaterials are being used at an industrial scale to solve real world problems and realising the most effect.

There are also several very important emerging applications in the nearby mm-wave band: for example, tuneable metasurfaces and phased arrays are likely going to be important for phase control in the 6G bands [[13, 14]]. Finally, we point out that the principal demonstrated in our manuscript for microwave metasurfaces can also be applied for mm-wave and THz frequencies. To better clarify this broad appeal of our work, we have added an entirely new portion of the discussion section focused on minimally perturbative imaging of arbitrary metasurfaces.

REVIEWER 3

“In this study, Harry et al. introduce a novel technique for characterizing the near-field electromagnetic properties of metamaterials. The experimental results underscore the method’s high sensitivity and resolution, effectively unveiling the specific effects of various defects on the resonant states of meta-atoms. This approach offers an innovative and potent experimental tool for the assessment and analysis of metamaterials’ electromagnetic characteristics. Overall, my assessment is positive. However, I have some comments and suggestions where further clarification and improvement could enhance the manuscript:”

R3.1: “Compared to the work in references [32-33], the introduction of single-pixel imaging methods is one of the innovative aspects of the work. However, the main text and supplementary materials do not provide a clear elucidation of the implementation methods and system structure for the employed single-pixel imaging technique. The authors are advised to provide a more detailed introduction, including additional textual descriptions, structural diagrams, photos of the actual setup, or experimental photos, etc.”

Response: This is an important observation, shared also by reviewer 2 (R2.1). We have now added these requested details across the main text and supplementary material.

R3.2: “Further details on the structure of the excitation light source system are suggested. Furthermore, how does the system ensure uniform light intensity across various measurement positions within the illuminated area? Additionally, could the structure of the lighting system potentially interfere with the electromagnetic field of the metamaterials under test?”

Response: We thank the referee for pointing this out. We have now added images showing the uniformity of the microwave and optical beams (Fig. RS2) and a photograph of the experimental setup (Fig. RS1) to the supplementary and expanded our discussion of the imaging method in the main text. Visible illumination uniformity is achieved by using a light integration pipe. The components of the lighting system are sufficiently separated from the microwave beam as shown in new Fig. RS1, so do not cause issues.

R3.3: “What are the main considerations for the TM microwave transmission at an angle of 35° to the normal of the metasurface? Does this requirement restrict the applicability of this method to certain types of metamaterials, such as those sensitive to the angle of incidence?”

Response: We apologize for not making this clear in the original manuscript, and now clarify the reason in the main text. The meta-atoms in this structure support a well localized resonance and so exhibit minimal dispersion (TE and TM are also comparable). As the spectral response does not depend heavily on angle, we chose an off-normal angle to facilitate ease of photoillumination as shown in Fig. R2 and RS1. However, a similar approach can be used with any angle of incidence, reflection, transmission, scattering or polarization. For example, to measure at normal incidence one can use a dichroic beamsplitter such as ITO to enable collinear illumination of the microwave and optical beams.

R3.4: “Mapping results at different frequencies to the visible spectrum is a creative and intriguing method of display. However, as evident in Figure S9, the impact of varying frequency sampling is significant. Does the choice of frequency sampling interval affect the accuracy and reliability of diagnostic results across various meta-atom areas, as well as the complementary trends observed in images at nearby frequencies? Is there a foundational basis for selecting the frequency sampling intervals?”

Response: The spectral sampling needs to be sufficient in order to average over high frequency artifacts common to microwave measurement approaches. To address the referee’s point, we have now expanded our discussion on the impact of sampling:

In Fig. RS12(c) we also see that our imaging method is also able to detect the spatial variations in signal that arise from the presence of experimental artefacts such as standing waves, manifesting as broad vertical stripes in the images (which are typically observed as spurious rapid oscillations in conventional VNA spectra). This vertical banding may also be observed in the monochromatic images of figures RS13 and RS14 (in particular Fig. RS14 14.7-14.9 GHz). At nearby imaging frequencies (~100 MHz, comparable to the period of standing wave oscillation), this can give rise to the appearance of complementary regions similar to those in Fig. M2(b) and (c) of the main text but these are distinct in spatial character and frequency spacing from the defects we discuss in this work. In our hyperspectral images these artefacts will be effectively averaged out as long as spectral sampling of the imaging is sufficient as shown in Fig. RS12(d).

-
- [1] Rayko Ivanov Stantchev, Baoqing Sun, Sam M. Hornett, Peter A. Hobson, Graham M. Gibson, Miles J. Padgett, and Euan Hendry, “Noninvasive, near-field terahertz imaging of hidden objects using a single-pixel detector,” *Science Advances* **2** (2016), 10.1126/sciadv.1600190.
 - [2] Rayko I. Stantchev, David B. Phillips, Peter Hobson, Samuel M. Hornett, Miles J. Padgett, and Euan Hendry, “Compressed sensing with near-field thz radiation,” *Optica* **4**, 989 (2017).
 - [3] I. R. Hooper, E. Khorani, X. Romain, L. E. Barr, T. Niewelt, S. Saxena, A. Wratten, N. E. Grant, J. D. Murphy, and E. Hendry, “Engineering the carrier lifetime and switching speed in si-based mm-wave photomodulators,” *Journal of Applied Physics* **132**, 233102 (2022).
 - [4] Sichao Qu, Yuxiao Hou, and Ping Sheng, “Conceptual-based design of an ultrabroadband microwave metamaterial absorber,” *Proceedings of the National Academy of Sciences* **118** (2021), 10.1073/pnas.2110490118.
 - [5] Ting Shi, Lei Jin, Lei Han, Ming Chun Tang, He Xiu Xu, and Cheng Wei Qiu, “Dispersion-engineered, broadband, wide-angle, polarization-independent microwave metamaterial absorber,” *IEEE Transactions on Antennas and Propagation* **69**, 229–238 (2021).
 - [6] Sajjad Taravati and George V. Eleftheriades, “Microwave space-time-modulated metasurfaces,” *ACS Photonics* **9**, 305–318 (2022).
 - [7] Jun Chen Ke, Jun Yan Dai, Jun Wei Zhang, Zhanye Chen, Ming Zheng Chen, Yunfeng Lu, Lei Zhang, Li Wang, Qun Yan Zhou, Long Li, Jin Shan Ding, Qiang Cheng, and Tie Jun Cui, “Frequency-modulated continuous waves controlled by space-time-coding metasurface with nonlinearly periodic phases,” *Light: Science Applications* **11**, 273 (2022).
 - [8] Xuchen Wang, Mohammad Sajjad Mirmoosa, Viktor S. Asadchy, Carsten Rockstuhl, Shanhui Fan, and Sergei A. Tretyakov, “Metasurface-based realization of photonic time crystals,” *Science Advances* **9** (2023), 10.1126/sciadv.adg7541.
 - [9] Hady Moussa, Gengyu Xu, Shixiong Yin, Emanuele Galiffi, Younes Ra’di, and Andrea Alù, “Observation of temporal reflection and broadband frequency translation at photonic time interfaces,” *Nature Physics* **19**, 863–868 (2023).
 - [10] Maxim V. Gorkunov, Alexander A. Antonov, Vladimir R. Tuz, Anton S. Kupriianov, and Yuri S. Kivshar, “Bound states in the continuum underpin near-lossless maximum chirality in dielectric metasurfaces,” *Advanced Optical Materials* **9** (2021), 10.1002/adom.202100797.
 - [11] Thomas CaiWei Tan, Yogesh Kumar Srivastava, Rajour Tanyi Ako, Wenhao Wang, Madhu Bhaskaran, Sharath Sriram, Ibraheem Al-Naib, Eric Plum, and Ranjan Singh, “Active control of nanodielectric-induced thz quasi-bic in flexible metasurfaces: A platform for modulation and sensing,” *Advanced Materials* **33** (2021), 10.1002/adma.202100836.
 - [12] Vasileios G. Ataloglou, Michael Chen, Minseok Kim, and George V. Eleftheriades, “Microwave huygens’ metasurfaces: Fundamentals and applications,” *IEEE Journal of Microwaves* **1**, 374–388 (2021).
 - [13] Yingjie Jay Guo and Richard W Ziolkowski, *Advanced antenna array engineering for 6G and beyond wireless communica-*

tions (John Wiley & Sons, 2021).

- [14] Younes Ra'di, Nikita Nefedkin, Petar Popovski, and Andrea Alù, “Metasurfaces for next-generation wireless communication systems,” *National Science Review* **10** (2023), 10.1093/nsr/nwad140.

Response letter Hyperspectral imaging of microwave metasurfaces with deeply subwavelength resolution

We once again thank all the reviewers for their time and valuable feedback, which has certainly enhanced our manuscript. Our responses to the remaining comments are included below in blue text.

REVIEWERS' COMMENTS

Reviewer #1 (Remarks to the Author):

The authors satisfactorily addressed most of the concerns raised by the reviewers. There exists one remaining problem in the revised manuscript. "Spatial resolution beyond $\lambda/600$ " is a claim highlighted in the Abstract, however, the supporting data figure is provided in the Supplement Information. I strongly recommend the authors to move or copy the key figures to the main text to provide strong support to this important claim.

As recommended the image supporting this claim has now been moved to Fig. 6(b) of the main text. We thank the reviewer for their time and comments.

Reviewer #2 (Remarks to the Author):

With the recent revisions to the manuscript, I now have a clearer understanding of the broader impact of their work across the microwave, millimeter-wave, and THz bands. The study is sound, and I now believe that it deserves publication in Nature Communications.

We thank the reviewer for their time and comments.

Reviewer #3 (Remarks to the Author):

The authors refer to Fig. RS12(c)(d) in their response to R3.4, but these subfigures are not included in the Supplementary Material. However, all other concerns raised in my initial review have been adequately addressed, and I have no further comments to add.

With apologies, this seems to have been a typo, which should have referred to Fig. RS9(c),(d)., which impacted only the review response document. Therefore as no action is needed, we thank the reviewer for their time and comments.